# Type I IFN in Glomerular Disease: Scarring beyond the STING

**DOI:** 10.3390/ijms25052497

**Published:** 2024-02-21

**Authors:** Alexis Paulina Jimenez-Uribe, Steve Mangos, Eunsil Hahm

**Affiliations:** Department of Internal Medicine, Division of Nephrology, Rush University Medical Center, Chicago, IL 60612, USA; alexispaulina_jimenez-uribe@rush.edu (A.P.J.-U.); steve_mangos@rush.edu (S.M.)

**Keywords:** glomerular diseases, IFN-I pathway, intracellular pattern-recognition receptors, STING, sterile inflammation

## Abstract

The field of nephrology has recently directed a considerable amount of attention towards the stimulator of interferon genes (STING) molecule since it appears to be a potent driver of chronic kidney disease (CKD). STING and its activator, the cyclic GMP-AMP synthase (cGAS), along with intracellular RIG-like receptors (RLRs) and toll-like receptors (TLRs), are potent inducers of type I interferon (IFN-I) expression. These cytokines have been long recognized as part of the mechanism used by the innate immune system to battle viral infections; however, their involvement in sterile inflammation remains unclear. Mounting evidence pointing to the involvement of the IFN-I pathway in sterile kidney inflammation provides potential insights into the complex interplay between the innate immune system and damage to the most sensitive segment of the nephron, the glomerulus. The STING pathway is often cited as one cause of renal disease not attributed to viral infections. Instead, this pathway can recognize and signal in response to host-derived nucleic acids, which are also recognized by RLRs and TLRs. It is still unclear, however, whether the development of renal diseases depends on subsequent IFN-I induction or other processes involved. This review aims to explore the main endogenous inducers of IFN-I in glomerular cells, to discuss what effects autocrine and paracrine signaling have on IFN-I induction, and to identify the pathways that are implicated in the development of glomerular damage.

## 1. Introduction

Over one hundred years ago, Paul Ehrlich introduced the term “horror autotoxicus” to the field of immunopathology. He used this term to refer to the process whereby the immune system directs its attention to attack the host rather than protect it from foreign harmful agents. This term finds increased relevance today and can be used to describe many types of hypersensitivity reactions, including autoimmune diseases (part of the type II and III hypersensitivity reactions) and disorders stemming from a hyperinflammatory response, such as cytokine storm syndrome and other autoinflammatory diseases that damage the kidney.

Recently, molecules typically associated with an antiviral immune response, cyclic GMP-AMP synthase (cGAS) and stimulator of interferon genes (STING), have been linked to the development of chronic kidney disease (CKD) in the absence of infection. Instead, cGAS/STING activation is through host-derived endogenous nucleic acids. Most relevant, STING blockade results in reduced kidney damage [1], especially at the level of the glomerulus [2,3]. Interestingly, although the expression of type I interferons (IFN-I) is one of the most expected outcomes after the activation of cGAS/STING and other intracellular pattern-recognition receptors (PRRs), it is still not clear if the renoprotective effects of cGAS/STING blockade rely on the abrogation of IFN-I production more so than other mechanisms.

The IFN-I pathway has been extensively studied as part of the innate immune response against viral infections. Signaling begins with the recognition of viral or endogenous nucleic acids by cGAS-like receptors (cGLRs), RIG-like receptors (RLRs), or toll-like receptors (TLRs), all members of the intracellular pattern-recognition receptor family. Once this recognition occurs, IFN-I expression is induced and acts in an autocrine or paracrine fashion. Ultimately, several genes are expressed that reduce transcription and translation rates and alter nucleic acid and lipid metabolism to block viral replication [4]. However, the function of this pathway in sterile inflammation is not yet fully understood. With the description more than a decade ago of interferonopathies, a group of diseases characterized by overactivation of the IFN-I pathway, much attention turned to focus on the relationship between IFN-I and autoinflammatory effects [5,6]. New light was shed on how endogenous stress, through self-nucleic acid recognition, drives sterile inflammation.

Glomerular disease (GD) represents a group of immune-mediated diseases characterized by inflammation and fibrosis within the glomerulus causing well-described histopathological lesions that encompass: (1) membranoproliferative glomerulonephritis (MPGN), (2) minimal change disease (MCD), (3) focal segmental glomerulosclerosis (FSGS), (4) membranous nephropathy (MN), (5) IgA nephropathy (IgAN), (6) crescentic glomerulonephritis (CGN), and (7) lupus nephritis (LN) [7]. Although the current classification is helpful for diagnosis, several and often different etiologies are involved in the development of each. Indeed, among the different etiologies, infections are responsible for only a fraction of GD cases. Most are caused by auto-antibodies against a broad spectrum of glomerular cell-specific molecules, genetic defects leading to autoinflammatory disorders, or as a consequence of other non-communicable diseases such as diabetes [8], making many GDs part of the aforementioned “horror autotoxicus”.

Among the different molecular mechanisms involved in the pathogenesis of GDs, there is accumulating evidence that the IFN-I pathway participates in the progression of glomerular damage. Much evidence stems from patient data derived from kidney biopsies, identifying the upregulation of gene products involved in the IFN-I pathway. These patient-derived data are summarized in Table 1.

The most compelling evidence suggesting the direct involvement of the IFN-I pathway in the development and progression of GD comes from patients diagnosed with multiple sclerosis, hepatitis C, and some types of cancer, all developing glomerular lesions after receiving recombinant IFN therapy [16,17,18,19]. Moreover, patients with inborn errors causing enhanced IFN-I production such as Aicardi–Goutières syndrome [20], STING-associated vasculopathy with onset in infancy (SAVI) [21,22], and systemic lupus erythematosus (SLE) [23] also present with glomerular lesions.

Results from experiments using animal models have provided additional support for the direct effect of IFN-I on the glomerulus. A murine model of nephritis, generated by injecting anti-glomerular basement membrane (GBM) antibodies, showed an increase in IFNα in renal tissue and a correlation with renal dysfunction demonstrated by increased levels of blood urea nitrogen (BUN) and proteinuria. Moreover, the exogenous overexpression of IFNα using an adenoviral vector increased proteinuria and led to more glomerular damage in treated animals [24]. In a similar approach, Migliorini et al. combined a model of adriamycin nephropathy with the administration of IFNα and IFNβ and showed enhanced kidney dysfunction through the perturbation of parietal epithelial cells (PEC) and podocyte loss [25]. In testing a novel model of murine SLE, Nacionales et al. showed that by blocking IFN-I activity by knocking out an IFN-I receptor (IFNAR), kidney dysfunction and glomerular injury could be ameliorated [26]. Taken together, results from these experiments suggest that the different causes of GD drive a diverse set of complex immunopathologies, including the stimulation of the IFN-I pathway, which, in turn, participates in further aggravating the disease.

The goal of this review is to (1) present a general overview of the IFN-pathway; (2) explore how the IFN-I pathway is induced, focusing on endogenous stress activators such as host-derived self-nucleic acids; and (3) outline how kidney-specific cells are affected by these cytokines.

## 2. IFN-I Induction in Sterile Inflammation

An early immune response to foreign pathogens, such as viruses, is the activation of PRRs, leading to the induction of IFN-I expression. This response can also be triggered by endogenous danger signals, mainly self-nucleic acids detected in the cytosol [27], but also by proteotoxic stress [28] and elevated cellular oxidative stress [29]. In physiological homeostatic conditions, mechanisms are in place to avoid aberrant PRR activation by self-nucleic acids. These include quick degradation using DNA and RNA nucleases, protective epigenetic and posttranscriptional modifications, and physically compartmentalizing the nucleus and mitochondria. However, nucleic acids can be damaged under certain stress-inducing conditions such as exposure to xenobiotics, metabolic stress, elevated reactive oxygen species, and the unfolded protein response. Nucleic acid damage can lead to their release and accumulation in the cytoplasm in the form of free cytosolic nucleic acids, micronuclei, chromatin fragments, or transposable elements [30,31], as illustrated in Figure 1.

In many cases of CKD, including several types of GD, patient biopsies show evidence of nucleic acid damage [32]. In a retrospective study using human biopsy samples, Ott et al. observed DNA fragmentation across a spectrum of different types of GD [33]. More specifically, in mice, it was shown that double-stranded DNA breaks in podocytes caused by the endonuclease I-Ppol correlated with proteinuria, glomerulosclerosis, and tubulointerstitial fibrosis [34]. This study also identified portions of the immune system as being responsible for the development of disease. Similarly, Dhillon et al. show that the increased levels of transposable elements (TE) and retrotransposons associated with DNA damage were detected in diseased samples from human patients and mouse models of renal fibrosis, suggesting a link with IFN-I pathway activation [35]. In support of these findings, De Cecco et al. showed an association between retrotransposable elements and IFN-I pathway signaling through the detection of cytoplasmic cDNA [36].

As previously mentioned, the first response in avoiding intracellular PRR activation is to degrade nucleic acids. Mechanisms to guard against immune activation caused by self-nucleic acid detection exist in the kidney. For example, case study data show a link between mutations in TREX1 and RNaseH2, key nucleases involved in cytoplasmic nucleic acid degradation, and thrombotic microangiopathy (TMA) or Aicardi–Goutières syndrome with kidney alterations [20,37], respectively.

If self-derived nucleic acids accumulate in the cytoplasm due to their release from the nucleus or a lack of degradation, there is a vast variety of PRRs able to recognize them, including cGLRs, RLRs, and TLRs [27]. Table 2 provides a general summary of the distinct types of intracellular PRRs and their cognate ligands.

Although different nucleic acid receptors activate different signal transduction pathways, most converge at the nexus of activating TANK-binding kinase 1 (TBK1) and, in turn, the phosphorylation of interferon regulatory factors (IRF), transcription factors that allow IFN-I expression. Figure 1 provides a graphical representation of the main molecular interactions that occur to initiate and propagate signaling through the IFN-I pathway. The remainder of this review will highlight certain aspects of these interactions as they relate to the available data concerning glomerular-specific diseases and other prominent kidney diseases.

## 3. cGAS/STING Activation in the Kidney

cGAS and its analogs, cGAS-like receptors (cGLR), are evolutionary conserved receptors in metazoans that detect cytosolic double-stranded (ds) DNA [38]. Once these receptors recognize their ligands, they have the ability to enzymatically convert triphosphate nucleotides into signaling linear or cyclic dinucleotides. For example, cGAS can generate 2′3′-cGAMP, a second messenger that binds and activates STING in the endoplasmic reticulum [39,40,41]. STING then oligomerizes and translocates to the Golgi, leading to the activation of TBK1 and IRF3, which in turn translocates to the nucleus and induces IFN-I expression, as shown in Figure 1.

Primarily due to its cellular localization and inability to discriminate between host and foreign dsDNA, the cGAS/STING system could be the first mechanism to respond to nuclear and mitochondrial (mt)-derived DNA present in cytoplasm. Interest in the cGAS/STING pathway has recently increased within the field of nephrology with the discovery that key glomerular cells express components of the pathway at the early stage of kidney dysfunction. Zang et al. show that podocyte injury can be triggered in a mouse model of diabetic kidney disease (DKD) through lipotoxicity-induced mitochondrial damage and mtDNA leakage leading to activation of the cGAS/STING pathway [3]. Mitrofanova et al. observed similar results in cultured human and mouse podocytes, as well as in mouse models of DKD and Alport syndrome that were exposed to a STING-specific agonist [2]. In an attempt to identify molecular pathways responsible for apolipoprotein 1 (APOL1)-induced kidney dysfunction, Wu et al. identified the expression of STING as an early event leading to podocyte cytotoxicity and cell death [42]. In all cases, pharmacologic inhibition or genetic deletion of the STING pathway reduced glomerular damage in these mouse models of disease. A more detailed exploration of the involvement of the cGAS/STING pathway in the progression to LN end-stage renal disease (LN-ESRD) using cultured human podocytes was recently reported. Davis et al. showed that if podocytes are treated with nucleosome-associated dsDNA fragments, a common marker found in the blood of Lupus patients, IFNγ-inducible protein 16 (IFI16) triggers the expression of both IFNβ and APOL1 and the activation of the cGAS/STING pathway [43]. More directly, this group showed that exposing podocytes to IFNβ promotes the expression of IFI16, suggesting a positive feedback loop between dsDNA recognition and cGAS/STING pathway activation. Additional evidence implicating the cGAS/STING pathway in lupus-mediated podocyte damage was provided by Li et al., who treated mouse podocytes with LN patient serum. It was shown that in these podocytes, the cGAS/STING pathway is activated, IFN-I expression is stimulated, and endoplasmic reticulum stress is induced, leading to apoptosis [44]. Podocytes from diabetic mice also show increased levels of cGAS and STING as well as activation of TBK1. Palmitic acid, a fatty acid associated with DKD, can elicit mitochondrial damage and leakage of mtDNA into the cytoplasm to activate the STING pathway. As a result, several inflammatory markers are induced, and levels of apoptosis increase. Interestingly, IFN-I expression does not seem to be elevated in this mouse model [3].

The bulk of research linking the activation of the cGAS/STING pathway to glomerular disease has focused on the podocyte; however, some evidence exists showing that other cells within the glomerulus are affected by this pathway and could contribute to renal dysfunction. There are several reports that implicate the STING pathway, either directly or indirectly, in glomerular endothelial cell (GEC) damage. In diabetic mice, Qi et al. present evidence that mtDNA contributes to glomerular damage indirectly through GECs by inducing podocyte loss, leading to proteinuria [45], while Caselena et al. show that GECs display severe mitochondrial damage when treated with high-glucose media [46]. Mitochondrial dysfunction is also seen in GECs from a transgenic mouse model of FSGS. This study showed crosstalk between glomerular cells, where the activation of transforming growth factor β receptor 1 (TGFβR1) in podocytes is associated with an increase in mitochondrial stress and release of mtDNA in GECs, leading to a subsequent loss of podocytes [47]. More indirect evidence comes from studies using mice expressing an APOL1 renal risk variant (RSK). The endothelial cells of these mice display high levels of mitophagy with mtDNA leakage into the cytoplasm and IFN-I pathway activation [48]. Although mostly indirect, there is evidence that the cGAS/STING pathway is activated in GECs and that this activation leads to glomerular damage. It is interesting to note that in 2009, Hagele et al. reported that GECs could be activated by dsDNA in a TLR-independent manner, showing increased IFN-I pathway expression [49]; this report was published prior to the cGAS/STING pathway being discovered [50,51].

Activation of the cGAS/STING pathway in mesangial cells (MCs) is still somewhat unexplored. There is some evidence, however, that mitochondrial damage leading to leakage of mtDNA occurs in cultured human mesangial cells (HMCs). This group showed that culturing HMCs with galactose deficient IgA from IgAN patients leads to mitochondrial damage due to the reduction in peroxisome proliferator-activated receptor alpha (PPARα) expression [52]. Curiously, MCs grown in hyperglycemic conditions have also shown markers of mitochondrial damage, suggesting that leakage of mtDNA and the subsequent activation of the cGAS/STING pathway could also be operating in some instances of DN.

It should be included here that there are other cell types not typically associated with the glomerulus that could also contribute to glomerular injury through the STING pathway. As an example, plasmacytoid dendritic cells (pDCs) are a special type of immune cell characterized, in part, by their high expression of IFN-I. Indeed, Deng et al. showed that pDCs rapidly infiltrate the kidney after AKI and produce IFNα, leading to kidney damage [53]. These cells have also been implicated in LN [54,55], where it has been reported that STING activation promotes the maturation of pDCs with their subsequent participation in glomerular injury [56]. However, it must be emphasized that it is still not clear if pDCs’ contribution to renal disease is through their involvement systemically or at the level of the kidney.

## 4. RLR Activation in the Kidney

Melanoma differentiation-associated gene 5 (MDA5) and retinoic acid-inducible gene I (RIG-I) are two RLR family members that reside in the cytoplasm, act as receptors for dsRNA, and, in a manner similar to cGAS, activate the mitochondrial antiviral-signaling protein (MAVS), TBK1, and IRF3 to elicit IFN-I expression. Although cytoplasmic endogenous dsRNA is a less common indicator of cellular stress as compared to cytoplasmic dsDNA, sources include retrotransposons, mt-dsRNA, and other secondary RNA structures that arise from epigenetic alterations, mitochondrial damage, or defects in RNA processing [57,58,59,60,61,62]. Murine models of autoimmune disease have been instrumental in showing that mt-dsRNA plays a relevant role in driving sterile inflammation through IFN-I pathway activation [63,64].

The activation of RLRs has been documented in a variety of kidney diseases, both in human patient samples and animal models of AKI and several different types of GD. A number of studies have shown that tubule-specific damage is linked to RLR activation. For example, Zhu et al. showed that renal tubule injury in mice caused by ischemia-reperfusion injury (IRI) or unilateral ureteral obstruction was closely tied to an accumulation of mt-dsRNA [65]. Similar results were reported by Doke et al., also using the murine IRI method [66]. Exploring the mechanisms responsible for AKI caused by crush syndrome (CS), Wang et al. utilized rats and were able to show a significant increase in RIG-I-associated signaling [67]. A study using whole-exome sequencing in LN patients identified a gain-of-function mutation in RIG-I leading to its constitutive activation [68], and in a separate study using kidney samples from CKD patients, RLR activation was associated with increased levels of retrotransposons [35].

Cells within the glomerulus also express the major sensors for dsRNA and can activate the associated immune signaling pathways. Yamashita et al. used cultured human and mouse podocytes to show that regardless of the source of dsRNA, be it from mitochondria, retrotransposons, or secondary structures spontaneously formed, RLR activation in podocytes leads to the expression of IFN-I, IL-6, and cytoskeleton alterations [69], the latter being instrumental in causing effacement and eventual cell loss [70]. The remaining bulk of evidence linking RLRs to podocyte dysfunction comes from investigations into the mechanisms whereby APOL1 causes kidney damage. Fang et al. used long-term injections of recombinant APOL1 in mice and observed increased expression of RIG-I in podocytes. Similar findings were seen in cultured human podocytes engineered to express APOL1 RSK. Knock-down of RIG-I, either using siRNA against RIG-I in in vitro experiments or adeno-associated virus short harpins (AAV-sh) in in vivo mouse experiments, blunted the expression of several pro-inflammatory genes and attenuated podocyte and glomerular damage, respectively [71]. One caveat emerging from this study is that it remains unclear what the direct activator of RIG-I is, and if RIG-I drives the expression of IFN-I along with other proinflammatory genes. Further confounding the involvement of RLRs in APOL1-mediated podocyte damage is the discovery that APOL1 mRNA itself possesses structural characteristics that allow for the formation of dsRNA secondary structures [72], which indeed have been shown to be recognized by MDA5 in podocytes [73]. However, as just mentioned, it is yet to be determined if these APOL1 secondary structures can be also recognized by RIG-I.

There is a dearth of evidence for the expression of RLRs in other glomerular cells. Results from in vitro experiments support the notion that RLRs can contribute to glomerular injury. Hagele et al. used a synthetic agonist of RLR to treat cultured murine GECs. These experiments revealed that there is an increase in the expression of IFN-I, IL-6, and intracellular adhesion molecule 1 (ICAM-I). It can be extrapolated that these changes in GECs could lead to increased albumin permeability [74]. The same synthetic agonist was used to treat MCs in vitro. The result was an increase in the expression of RIG-I chemokine ligand 5 (CCL5) and the C-X-C motif chemokine ligand 10 (CXCL10) [75,76]. Taken together, these results suggest that if GECs and MCs are under stress and exposed to RLR ligands, they are able to promote inflammation, favoring chemotaxis and the adhesion of leukocytes. However, since the synthetic ligand for RLRs can also activate TLR3, signaling through TLR3-mediated pathways cannot be ruled out.

## 5. TLR Activation in the Kidney

TLRs comprise a large family of PRRs that are expressed on the plasma membrane and intracellularly in endosomes, the latter location being activated by nucleic acids and highly associated with sterile inflammation [77]. Endosomally restricted TLRs include: TLR3 (which recognizes dsRNA), TLR7/8 (which recognizes ssRNA), and TLR9 (which recognizes dsDNA). It must be highlighted that intracellular TLRs cannot participate in the first response to self-derived nucleic acids in the cytoplasm as the other RLRs mentioned above can since they are physically sequestered within endosomes. Instead, these TLRs are only activated in the event that extracellular DNA or RNA from damaged or dead bystander cells has been endocytosed and the nucleic-acid-containing endosome fuses with an acidified endosome containing the TLRs [78]. Indeed, there are several reports in the literature where extracellular mRNA or mitochondrial RNA derived from damaged cells has been engulfed by neighboring cells, resulting in TLR3 activation and sterile inflammation [63,79,80]. Highlighting a route for entry for nucleic acids, Bertheloot et al. showed that RNA released from damaged cells can be detected by the receptor for advanced glycation end products (RAGE) to elicit their internalization and activate TLR7 and TLR8 [81]. It is thought that a similar mechanism exists for the detection and internalization of mtDNA from damaged mitochondria, leading the activation of TLR9 [82]. Extracellular nucleic acids are shielded from degradation if they are bound to other molecules. These shielding molecules include antibodies, antimicrobial peptides, neutrophil extracellular traps (NETS), and microvesicles, all of which have been detected in patients with lupus [83,84,85,86]. Once formed, these nucleic acid–protein complexes can be taken up through endocytosis by specialized cells, including macrophages and dendritic cells, but also by podocytes, MCs, and GECs [87,88,89]. However, we should consider that the activation of this class of TLRs could be the consequence of previous IFN-I or other pathway activation and/or as a result of actions carried out by any number of immune cell types, rather than directly by cells within the glomerulus. Interestingly, some RNA-based prophylactics and therapeutics have been associated with altered kidney function [90,91,92], suggesting that these exogenous nucleic acids could be implicated in the induction of the IFN-I pathway. On the other hand, there is also experimental evidence pointing to the advantages of RNA-based drugs that target coding and non-coding RNAs associated with the development of kidney disease [93,94].

The function of TLRs in CKD and GD has been extensively reviewed in previous publications [95,96]; therefore, this review will narrowly focus on the involvement of TLR signaling in podocytes, GECs, and MCs.

Experiments using cultured human podocytes confirm that these cells highly express TLR3 and, upon its activation, were shown to induce the expression of IFNβ, several chemokines (IL-6, MCP-1, CCL5), the costimulatory molecule CD80, and APOL1, but not IFNα [97,98,99]. TLR7 agonists have been widely used to generate mouse models of lupus, which display varying degrees of podocyte injury; however, podocyte damage can be traced to the action of other cell types, including those that belong to the immune system [100,101]. A more direct involvement of TLR8 in podocyte damage has been seen in mice models of lupus and in rodents that have undergone unilateral ureteral obstruction. In these experimental models, TLR8 expression is increased in podocytes and is associated with the elevation of micro-RNA 21 (miR21) [102,103], an endogenous ligand for TLR8 [104]. Interestingly, it has been reported that miR21 is increased in the urine and renal tissue of CKD patients, including those diagnosed with DN and IgAN [105,106,107]. In a rat model of nephrotic syndrome and in a mouse model of autoimmune GN, TLR9 expression is increased and associated with podocyte injury [108,109]. Moreover, mtDNA and/or dsDNA seems to be the ligand for this receptor in podocytes, inducing their apoptosis in vitro. Seemingly contradictory results were reported by Bossaller et al., who showed that TLR9 deficiency aggravates LN induced by TLR7 agonists in mice [110], suggesting a role for TLR9 as a negative regulator to limit glomerular damage.

Only the involvement of TLR3 in GECs has been reported. A number of publications originating from the same group using cultured human cells showed that the in vitro activation of TLR3 elicits the expression of IFN-I, RLRs, chemokines, and plasminogen activator inhibitor-1 (PAI-1), a negative regulator of fibrinolysis [111,112,113,114,115,116]. Taken together, this body of work suggests that the activation of TLR3 in GECs may contribute to inflammatory and fibrotic responses, events that could also be a result of a secondary response to the IFN-I pathway signaling.

Similar to reports for GECs, much of the work exploring the action of TLRs in MCs was accomplished using cultured human mesangial cells. These studies show that the in vitro activation of TLR3 in MCs results in the expression of several different chemokines, adhesion molecules, and matrix metalloproteinase 9 [117,118,119,120]. It is worth mentioning that this metalloproteinase is an enzyme relevant to the inflammatory response and is detected in the biopsies of patients diagnosed with active and chronic GD [121], as well as in the urine of patients with MN [122]. A report by Shen et al. revealed that TLR9 was significantly elevated in MCs grown in vitro under hyperglycemic conditions and that knockdown of TLR9 under these same conditions reduces the expression of several inflammatory markers and the incidence of apoptosis [123]. In the same study using a mouse model of diabetes, silencing TLR9 reduced glomerular matrix cell expansion.

Taken together, these various reports suggest that under conditions that result in cellular stress, nucleic acids may be released into the extracellular space, where they can be taken up by any number of different cells, including cells within the glomerulus. This, in turn, results in the activation of the IFN-I pathway and may include other signal pathways, such those that induce nuclear factor κB (NF-κB) activation. The end result is to prolong and propagate sterile inflammation in the kidney.

## 6. Effect of IFN-I in Glomerular Cells

As described above, the activation of intracellular PRRs leads to the synthesis of IFN-I. The term ‘IFN’ encompasses a group of cytokines divided into three family types referred as type I, type II (also named γ), and type III (also named λ). Interferons were initially discovered for their essential role in the anti-viral and anti-cancer immune response, and more recently, for their participation in autoimmune and autoinflammatory disorders [6,124,125,126]. Among the IFN-I family members, several subtypes have been described in mammals, including α, β, δ, ε, κ, τ, and ω. In humans, IFN-I contains 13 isoforms of IFNα and one isoform of IFNβ, and they are primarily recognized by their biological function [124,127].

Type IFN-Is are recognized by a heterodimeric receptor composed by interferon α/β receptor subunits 1 and 2 (IFNAR1 and IFNAR2). Although both subunits are necessary for IFN recognition, the signal transduction is initiated by IFNAR1. Attached to the cytoplasmic tail of IFNAR1 is tyrosine kinase 2 (TYK2), which is initially activated, followed by the phosphorylation of Janus kinase 1 (JAK1) and the phosphorylation of signal transduction and activator of transcription (STAT)1 and STAT2, which are associated with the cytoplasmic tail of IFNAR2. Next is the formation of ISGF3, a heterotrimeric transcriptional factor made up of STAT1, STAT2, and IRF9, which functions to promote the expression of interferon-stimulated genes (ISGs), as shown Figure 1. Cellular stress, including ribosomal stress, drives crosstalk between the above-mentioned canonical IFN-I pathway and other signaling pathways to boost the expression of hundreds of ISGs. The end result is the alteration of several cellular processes, such as decreased transcription due to RNA degradation triggered by ISG20, a diminished translation rate due to protein kinase R activity, and changes in lipid metabolism by cholesterol 25-hydroxylase [4,128,129,130].

In mice, IFNAR1/2 have been reported to be ubiquitously expressed in all tissues, and after systemic activation with recombinant IFN-I injection, the kidney shows a strong response to these cytokines [131,132,133]. In human kidney cells, there is little information related to their expression levels. Chang et al. provide some peripheral evidence of IFNAR1 expression in para-tumoral healthy tissue from renal cell carcinoma biopsies [134]. Several glomerular diseases display enrichment of the IFN-I pathway (summarized in Table 1), strongly suggesting the activation of IFNAR1/2. However, there is a substantial lack of direct evidence related to their expression levels in different glomerular cells, either in normal or pathological conditions.

Recently, Manoharan et al. reported that IFNAR1 expression on podocytes, where it can physically interact with tissue factor (TF) under steady-state conditions due to its homology with IFNAR2, acts to restrict IFN-I pathway activation. Moreover, TF deletion in podocytes promotes IFN-I pathway activation and the development of glomerular damage. Interestingly, biopsies from patients diagnosed with different forms of GD show a reduction in TF levels within the glomerulus [135], suggesting that the loss of IFNAR1 and TF interaction could promote IFN-I pathway activity.

The effects of IFNα signaling in podocytes includes the induction of autophagy and the inhibition of the mammalian target of rapamycin complex 1 (mTORC1) pathway [136]. These observations provide an additional link between IFN-I signaling and the development of GD since the loss of mTORC1 in podocytes causes proteinuria associated with glomerular damage [137]. Furthermore, IFNβ induces the expression of APOL1 [99] and apoptosis in mature podocytes, along with the reduction of nephrin expression during podocyte differentiation in vitro [25]. Li et al. also showed that IFN-I activation leads to the induction of viperin (also known as RSAD2), a common ISG that has been reported to be a negative regulator of podocyte differentiation [138]. Similar effects have been described in other kidney-specific cells, such as parietal epithelial cells (PECs) and tubule epithelial cells (TECs), where IFN-I signaling disrupts the cell cycle and promotes cell death [25,139]. It is becoming clear that elevated IFN-I expression in podocytes can drive autophagy, impair cellular differentiation, and promote podocyte apoptosis, events that can exacerbate glomerular disease. On the other hand, Lee et al. showed that by targeting IFNAR1/2 signals using bariticinib, a JAK inhibitor, podocyte damage in a mouse LN model could be reduced, along with a general reduction in inflammation, pointing to a systemic effect of the inhibitor [140].

There is little direct evidence of what effect IFN-I has on glomerular endothelial cells. A series of publications from Hiroshi Tanaka’s group using cultured human GECs identified a set of ISGs with the potential to be used as biomarkers for LN [111,141,142]. Instead, we might draw on work accomplished using other types of endothelial cells to extrapolate IFN-I signaling effects on GECs. For example, work using brain, liver, and lung endothelial cells has indicated that IFN-I cytokines, mainly IFNβ, affect tight junctions through the down-regulation of VE-cadherin, reducing the synthesis of nitric oxide, the induction of PAI-1 secretion, and the reduction in caveolin 1 expression [143,144,145]. Combined, these effects could promote endothelial permeability and stiffness. Additionally, interferon-stimulated gene 15 (ISG15), one of the byproducts of IFNAR1/2 activation, has been linked to the development of vascular stiffness in cases of hypertension [146].

In MCs, it has been demonstrated that IFNβ promotes the expression of IL-6 and participates in regulating proliferation [147]. Using primary HMCs, Zhang et al. discovered a feed-forward regulatory pathway where IFN-I stimulates the expression of miR744, which boosts the activation of JAK1/STAT1/STAT2 to promote the expression of several chemokines, including CXCL10 [148]. More recently, Gao et al. showed that CXCL10 can stimulate mesangial cell proliferation and migration in vitro and participates in mesangial cell expansion in a mouse model of glomerulonephritis [149]. Figure 2 summarizes the key findings of this review with respect to the effect of the IFN-I pathway on glomerular cells. Many questions, however, remain unanswered. For example, is it possible that IFN-I pathway activation is merely an indirect reaction to an initial insult rather than the main cause of GD? Since GD is a group of diseases with each group displaying a complex pathophysiology, it may prove difficult to accurately uncover the responsible mechanisms for each since in vivo and in vitro models seldom fully resemble the pathology displayed by human patients. Nonetheless, and despite the complexity of the molecular mechanisms involved, it is clear that the IFN-I pathway is participating in the development and progression of GD.

## 7. Concluding Remarks and Future Perspectives

The evidence of IFN-I pathway function in glomerular cells presents an opportunity to explore the potential exploitation of this pathway for therapeutic purposes in GD. For example, manipulation of the IFN-I pathway and related cytokines has been reported to boost the production of antibodies in vaccination [150,151]. Therefore, in the context of autoimmune-associated GD, abrogating IFN-I pathway signaling could potentially have the opposite effect by decreasing the production of auto-antibodies. This idea is supported by a clinical trial that demonstrated the reduction of auto-antibodies in systemic lupus using a JAK1/2 inhibitor [152]. In a different study, blocking upstream of this pathway by targeting STING appears to be a promising strategy given recent progress in drug discovery [153].

Another avenue is to explore the effect of the IFN-I pathway on the metabolism of glomerular cells, including glycolysis, oxidative phosphorylation, and lipid synthesis, as has been reported for macrophages and T cells, affecting their function [154,155,156], or its relation with other cellular stress responses, such as the activation of inflammasomes and the unfolded protein response (UPR) [28,157]. Finally, new insights into renal fibrosis, glomerular cell senescence, and aging could be gained by exploring IFN-I pathway crosstalk with the transforming growth factor β (TGF-β) pathway [158], DNA damage and epigenetic changes [159], or the inflammaging process [160], respectively.

## Figures and Tables

**Figure 1 ijms-25-02497-f001:**
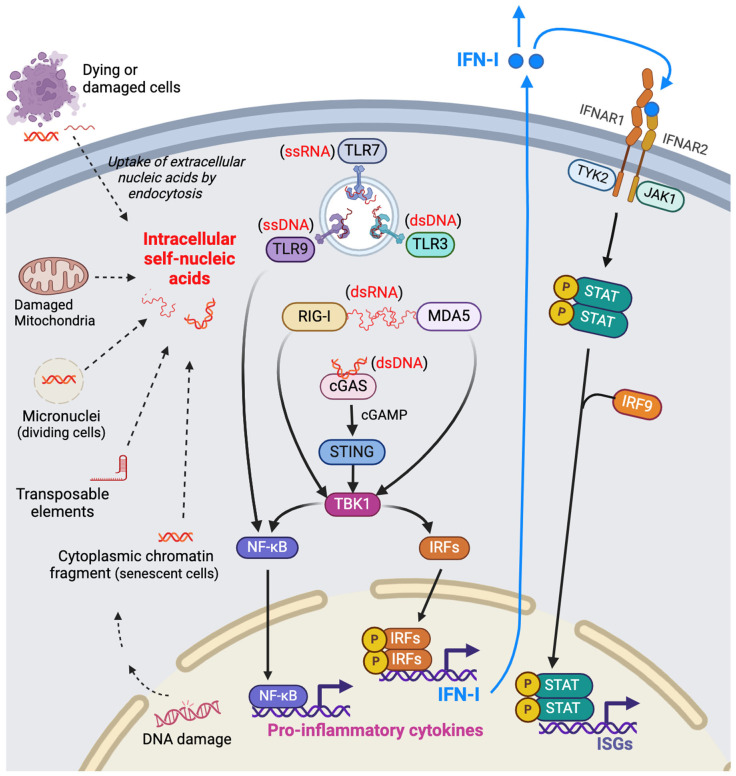
Origins and intracellular sensors of self-nucleic acids promoting the IFN-I signaling pathway. During cellular stress, self-derived nucleic acids (DNA or RNA) are released from damaged mitochondria and the nucleus, consequently accumulating in the cytosol. In addition, extracellular nucleic acids liberated from neighboring dying or damaged cells are internalized by endocytosis. These intracellular (cytosolic or endosomal) nucleic acids are recognized by diverse intracellular nucleic acid sensors, triggering the activation of the signaling pathways that produce IFN-I and proinflammatory cytokines. Specifically, sensors for endosomal nucleic acids include TLR3 (for double-stranded DNA), TLR7 (for single-stranded RNA), and TLR9 (for single-stranded DNA). Cytosolic double-stranded RNA is detected by RIG-I or MDA5, while cytosolic double-stranded DNA is recognized by members of the cGAS-STING pathway. Activation of these intracellular nucleic acid sensors stimulates TBK1 activation, prompting the translocation of IRFs and NF-kB into the nucleus. There, they orchestrate the expression of IFN-I and proinflammatory cytokine genes. Subsequently, binding of IFN-I to IFNAR1/IFNAR2 triggers the activation of the JAK-STAT pathway, culminating in the induction of ISGs. This cascade of events illustrates the intricate molecular mechanisms involved in the recognition of self-nucleic acids, the subsequent activation of signaling pathways leading to the expression and secretion of IFN-I and proinflammatory cytokines, and the subsequent induction of ISGs via the JAK-STAT pathway. This figure was created with BioRender.com. IFN-I, type I interferons; TLR, toll-like receptor; RIG-I, retinoic acid-inducible gene I; MDA5, melanoma differentiation-associated protein 5; cGAS, cyclic GMP-AMP (cGAMP) synthase; STING, stimulator of interferon genes; TBK1, TANK-binding kinase 1; IFNAR, type I interferon receptor; ISGs, interferon-stimulated genes.

**Figure 2 ijms-25-02497-f002:**
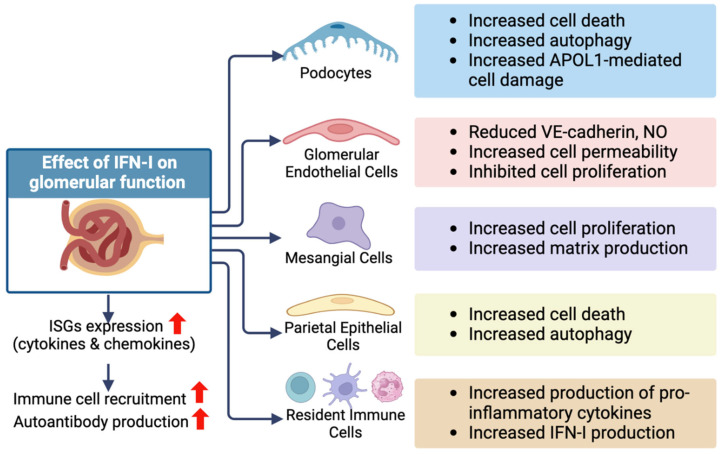
The potential deleterious effects of IFN-I on different glomerular cell types. IFN-I can exhibit a direct influence on diverse cell types within the glomerular compartment, altering their cellular functions. In addition, increased IFN-I levels induce the expression of interferon-stimulated genes (ISGs). This could intensify inflammatory processes by recruiting inflammatory immune cells to the kidney and increasing the production of autoantibodies. This figure was created with BioRender.com. APOL-1, apolipoprotein L1; VE-cadherin, vascular endothelial-cadherin; NO, nitric oxide. Red arrows indicate increased responses.

**Table 1 ijms-25-02497-t001:** IFN-I pathway components enriched in glomerular disease.

Glomerular Disease	Observation	Ref.
MPGN C3 glomerulopathy	Interferon-stimulated genes (ISG) IFI44L, IFIT1, MX1, and OAS2 enriched	[9]
MCD	Immunohistochemical detection of STING in the glomerulus ISG and PRR genes enriched (IFIH1, IRF7, OAS1, TLR7, TLR8)	[10] [11]
FSGS	ISG and PRR genes enriched (IFIH1, IRF7, OAS1, TLR7, TLR8) IFN-I signaling pathway enrichment in glomerular endothelial cells	[11] [12]
MN anti-PLA2R	IFN-I signaling pathway in glomerular mesangial cells (IFI16 gene enrichment)	[13]
IgAN	Immunohistochemical detection of STING in glomerulus, tubules, and interstitium, associated with urinary mitochondrial DNA	[10]
LN	IFN-I signaling pathway enrichment ISG genes enriched (IRF5, OAS1, MX2, DHX58, IFITM1, ADAR, GBP2)	[14] [15]

**Table 2 ijms-25-02497-t002:** Intracellular pattern recognition receptors (PRRs) involved in IFN-I induction.

Family	Name	Ligands	Cell Distribution	Source of Endogenous Activators
cGLRs	cGAS	dsDNA	Cytoplasm	Nuclear DNA Mitochondrial DNA
RLRs	RIG-I	dsRNA	Cytoplasm	Mitochondrial dsRNA Transposable elements (TE) in the form of Alu RNA, RNA:DNA duplex Misedited RNA with secondary structures
	MDA5	dsRNA	Cytoplasm	
TLRs	TLR3	dsRNA	Endosomes	Extracellular nucleic acids from damaged bystander cells
	TLR7	ssRNA		
	TLR8	ssRNA		
	TLR9	dsDNA		

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
