# Peer review of "Type I IFN in Glomerular Disease: Scarring beyond the STING"

_ijms, 2024, doi:10.3390/ijms25052497_

Round 1

Reviewer 1 Report

Comments and Suggestions for Authors

Although the subject is potentially interesting, the structure of this review is rather chaotic and difficult to follow.

The authors tend to start each paragraph with general statements, not always correct. E.g. categories of GN look rather suspicious when histopathological diagnoses (membranous nephropathy, membranoproliferative GN etc.) are put together with clinical entities like lupus. The latter can take the histopathological form of any five categories listed before, so such classification is misleading.

Then the abrupt switch into detailed data from multiple references takes place, without logical explanation of connections between them.

Even if cGLRs seem the major subject, other PRRs are also commented on, disturbing the balance between the main plot and collaterals.

Pictures and their legends are clear and logical. Conclusions are too long and inconclusive.

The last picture should be discussed earlier and may in fact constitute the core of the review, not the summary.

Comments on the Quality of English Language

There is a tendency towards long sentences. The meaning of them is sometimes difficult to follow. Repetitions occur.

Author Response

Comments from Reviewer 2:

R2-1: Although the subject is potentially interesting, the structure of this review is rather chaotic and difficult to follow.

We are sorry to hear that reviewer 2 found the structure of our paper chaotic and difficult to follow. We feel that the structure is logical and flows easily from a general consideration of the IFN-I pathway to more specific details concerning the impact on the kidney. Given the remarks from the other 3 reviewers, we believe others feel the same as we do.

R2-2: The authors tend to start each paragraph with general statements, not always correct. E.g. categories of GN look rather suspicious when histopathological diagnoses (membranous nephropathy, membranoproliferative GN etc.) are put together with clinical entities like lupus. The latter can take the histopathological form of any five categories listed before, so such classification is misleading.

To address this point, we rephrased relevant sections in the manuscript so that we are using the term glomerular diseases (GD) instead of glomerulonephritis. The categories we used in this section of the article are in line with previously published reports (see reference #7 - “Tinawi, M. Update on the etiology, classification, and management of glomerular diseases. Avicenna J Med 10, 61-67, 2020”). This publication states that lupus nephritis should be in its own category and could be subcategorized as minimal mesangial, focal, proliferative, membranous and advanced sclerosing. Also, in response to the above comment from reviewer 2, we removed the term ‘classification from the manuscript and referred to the different categories of GD based on those contained within reference #7.

R2-3: Then the abrupt switch into detailed data from multiple references takes place, without logical explanation of connections between them.

To address this comment from reviewer 2, we condensed and summarized the ‘detailed data’ from this section of the manuscript into a Table. The new Table 1 summarizes IFN-I pathway components that are enriched in glomerular disease.

R2-4: Even if cGLRs seem the major subject, other PRRs are also commented on, disturbing the balance between the main plot and collaterals.

We apologize for any confusion our writing may have caused reviewer 2. We believed that the title of our article made it clear that the aim of the review is to explore different mechanisms, beyond cGAS/STING, whereby the IFN-I pathway is induced in GD. Our handling of the subject is to highlight that other PRRs (including RLRs and TLRs) can detect endogenous nucleic acids and induce IFN-I expression. Therefore, they are part of the main plot and not collaterals. To avoid confusion, we rephrase the abstract (lines 11, 18-22) and introduction (lines 49-62).

R2-5: Pictures and their legends are clear and logical. Conclusions are too long and inconclusive.

We thank reviewer 2 for this comment. The conclusion has been revised and shortened.

R2-6: The last picture should be discussed earlier and may in fact constitute the core of the review, not the summary.

We have moved Figure 2 to section 6, earlier in the manuscript.

R2-7: There is a tendency towards long sentences. The meaning of them is sometimes difficult to follow. Repetitions occur.

We have identified some longer, run-on sentences and modified them to make our points clear to the reader.

Reviewer 2 Report

Comments and Suggestions for Authors

This is a review of the impact of the IFN-I pathway on the kidney, focusing on endogenous inducers of intracellular PRRs in the development and progression of glomerulonephritis. I enjoyed reading the review. It is well-written and well-structured. I have only a suggestion. There have been significant advancements in mRNA and ncRNA-based treatments, some of which have been linked to renal dysfunction. I suggest including information about these inducers and proposing strategies to reduce kidney injury.

Author Response

We thank reviewer 3 for this suggestion.  Although we feel that a consideration of current and potential treatments for patients with nephrotic disease caused by aberrant IFN-I signaling is an important topic, it is beyond the scope of our review. However, we have included a short section in the revised manuscript mentioning the impact of mRNA and ncRNA-based treatments, both in a positive and negative manner which can be found in lines 386-392 of the revised manuscript. Due to this addition, we have added approximately 7 new references to our literature cited section.

Reviewer 3 Report

Comments and Suggestions for Authors

This is a comprehensive and very informative review on the role of the interferon pathway in glomerular diseases. The review is also very timely as accumulating evidence suggests an important role of the anti-viral pathway and interferons in kidney disease. In general, this is a well written and organized review, that is easy to read. It provides an excellent overview with ample details and appropriate references. The figures and table provide a very well usable summary of key information.

There are a couple of comments and suggestions that could further improve the text.

 1, The figures are very helpful by providing a summary of key concepts. It is recommended that the authors add more references to Fig 1 to the text, since this is a great summary of the signalling pathway referenced in the text. Add a reference to the figure toSection 3 (around line 178) and to Section 6 (second paragraph).

Figure 2 on the other hand could be useful to refer to earlier, as it could be helpful in the section that discussed the effects of IFNs on glomeruli (section 6). In fact, the summarized description might be better in this section rather than the conclusion.

2, In section 6, it would be useful if the authors added a comment on the expression of interferon receptors in the various cells of the glomerulus; especially since there is little direct research available in these cells.  Are there data that these are under regulation or change during pathological states?

3, The first part of the concluding remarks would be more appropriate as part of the introduction, or an earlier section.

4, Some parts contain a lot of abbreviations. If possible, it would be good to reduce the number of these. Some abbreviations are repeated only a few times, maybe consider not using an abbreviation for these.

5, In section 6 line 395 the authors describe the effects of interferon as “altered cellular processes such as translation, transcription and metabolism”. This is very general and not very informative. It is recommended that the authors provide some more details on the effects of interferon and make this general statement more specific.  

6, There are a few sentences/ words that need attention, as listed below:

Line 17: (in the abstract): this sentence is not clear;

Line 74: received should be replaced with receiving;

Line 122: trough should be replaced with through;

Line 217: beginning of the line “the” is missing after within (within the glomerulus);

Line 220: the authors likely meant Transforming Growth Factor instead of tumor growth factor?

Line 239: including should be replaced by included.

Comments on the Quality of English Language

No issues with English, a few minor corrections needed (many listed above)

Author Response

R4-1: The figures are very helpful by providing a summary of key concepts. It is recommended that the authors add more references to Fig 1 to the text, since this is a great summary of the signaling pathway referenced in the text. Add a reference to the figure to Section 3 (around line 178) and to Section 6 (second paragraph).

We thank the reviewer for these comments and agree that more references to Figure 1 are needed throughout the text of the review. We have added references to the figure in several places in the revised manuscript, including the specific spots suggested by reviewer 4 (now line 202).

R4-2: Figure 2 on the other hand could be useful to refer to earlier, as it could be helpful in the section that discussed the effects of IFNs on glomeruli (section 6). In fact, the summarized description might be better in this section rather than the conclusion.

We again agree with this comment from reviewer 4 (a similar comment was made by reviewer 2, see above) and have moved Figure 2 to section 6 of the revised manuscript, including a summary of the key findings as displayed in the figure.

R4-3: In section 6, it would be useful if the authors added a comment on the expression of interferon receptors in the various cells of the glomerulus; especially since there is little direct research available in these cells.  Are there data that these are under regulation or change during pathological states?

We agree that this would be beneficial and have included additional information that refers to the specific expression of IFNAR1/2 receptors in the kidney which can be found in lines 475-485 of the revised manuscript.  This necessitates the addition of 4 new references to the References section.

R4-4: The first part of the concluding remarks would be more appropriate as part of the introduction, or an earlier section.

As noted above, we have relocated the first two paragraphs of the conclusion in section 7 to the introduction in section 1, lines 49-63.

R4-5: Some parts contain a lot of abbreviations. If possible, it would be good to reduce the number of these. Some abbreviations are repeated only a few times, maybe consider not using an abbreviation for these.

We removed several abbreviations such as ROS, UPR, ER, UUO, MMP, ERV, ERE.

R4-6: In section 6 line 395 the authors describe the effects of interferon as “altered cellular processes such as translation, transcription and metabolism”. This is very general and not very informative. It is recommended that the authors provide some more details on the effects of interferon and make this general statement more specific.  

We have expanded this section and included more specific details about what cellular processes and functions are affected, along with the responsible interferon stimulated genes. (lines 468-474)

R4-7: There are a few sentences/ words that need attention, as listed below:

Line 17: (in the abstract): this sentence is not clear;

This sentence has been rewritten in the revised manuscript. We believe the meaning is now clear.

Line 74: received should be replaced with receiving;

Corrected, now line 90

Line 122: trough should be replaced with through;

Corrected, now line 146

Line 217: beginning of the line “the” is missing after within (within the glomerulus);

Corrected, now line 243

Line 220: the authors likely meant Transforming Growth Factor instead of tumor growth factor?

Corrected to Transforming Growth Factor, now line 253.

Line 239: including should be replaced by included.

Corrected, now line 276

Round 2

Reviewer 1 Report

Comments and Suggestions for Authors

All concerns have been addressed properly